# OpenReview forum: "Get RICH or Die Scaling: Profitably Trading Inference Compute for Robustness"
_ICLR.cc/2026/Conference — ICLR 2026 Poster_

### Official Review · Reviewer_Xw6j · 2025-10-25

**Soundness:** 3
**Presentation:** 3
**Contribution:** 3
**Rating:** 6
**Confidence:** 4

**Summary:**

The authors present a hypothesis, the "Robustness from Inference Compute Hypothesis" (RICH), which posits that spending compute at inference time as an adversarial defense technique is more effective the more in-distribution the contents of the attack are. They then perform a robustness exploration on three versions of the LLaVA VLM trained with different robustness "strengths", and show that the experiments support this hypothesis.

**Strengths:**

## originality
The "RICH" hypothesis is novel as far as I know. That something being in-distribution improves performance is not at all surprising, but formulating robustness in this way is not something I had heard about before

## quality
The work appears to be of high quality. While the authors primarily study "only" three models from the LLaVA model family, they do a large number of experiments (plus, academic compute levels should not be punished imo).

## clarity
The paper is above-average in terms of clarity. The clear narrative around RICH, and the use of boxes to showcase results, both significantly add to the presentation of the paper.

## significance
While I don't consider this result groundbreaking (as mentioned before, that the model is more robust in-distribution is not surprising), it is still a compelling and interesting piece of work which I believe adds to the (currently limited) literature on VLM adversarial robustness scaling.

**Weaknesses:**

Not too many complaints.

1) It would be nice to have a study where the authors themselves perform adversarial training, and then talk about increasing adversarial robustness as a result of increasing amount of adversarial training, akin to the approach in [1], in order to talk about models of different robustness levels. Using three different off-the-shelf models and simply stating they have low/medium/high levels of robustness feels like a strong assumption, and it's hard to rule out other confounders from the fact that the different variants were trained with different techniques. Is there a reason the authors were not able to perform adversarial training themselves (was it simply a scope/compute limitation, or is there another reason)?

2) softening the language a bit around the RICH hypothesis "definitely being the correct explanation" and being supported by all the experiments (I'm particularly uncertain about whether Figure 3 supports it). Basically being a bit more humble with the language here, especially given a lack of experiments on frontier models.

3) doing just a little bit more work in cleaning up the presentation and situating the work in the literature (I'll put a list of what to do in the questions).

---

[1] https://arxiv.org/abs/2407.18213

**Questions:**

* To what extent is it reasonable to say that the three models studied represent low/medium/high amounts of robustness? Is there a way to make this more convincing?
* In Figure 3 (right), you show that increasing K improves robustness across models. But in line 359-361 (in the box), you seem to say that it only helps the already-more-robust model (Delta2). This seems wrong based on Figure 3 (right). It might seem true based on Figure 3 (left), but that's only because the attack saturates, so I don't think it's fair to say even based on that. Would you agree? Am I missing something?
* Figure 1: what happens if you repeat the whole brown text (not just the bit in the curly braces)? Why did you not choose to repeat the whole instruction?
* in Figure 2, why do you think that adding only a subset of the defenses (middle two columns) leads to *worse* performance (I would have expected it to be at least the same, if not better)?

## Suggestions for improved clarity
* Figure 1 caption: the bold text is not so clear what it's trying to say
* line 160: why is there no box around the hypothesis as there is around the questions? seems it should be a fancy box even.
* line 214: "If a clear security..." I'm not sure this sentence is true? Like I don't see how it is implied.
* line 215: what does "generalizing to a specification and adversarially OOD data" mean?
* all tables: remove vertical bars. See https://people.inf.ethz.ch/markusp/teaching/guides/guide-tables.pdf
* line 297: it's not clear where this question and where its answer come from
* table 1: what is the "up arrow" next too the "attacker loss" text?
* table 1: why is there (0.0) next to the Delta2 numbers?
* table 1: what happens if you don't pre-fill the model response? Please mention in the main text why you needed to do this in order to get it to work.
* 304: "provides the same robustness benefits from applying the security specification" I'm not sure what this means
* 305: model -> model's
* 415: traditional -> realistic

---

> ### Author Response · Authors · 2025-11-21
>
> Thank you for your insightful review! We are pleased that you found our presentation of the Robustness from Inference Compute Hypothesis (RICH)  to be “novel”, “of high quality”, and to have a “clear narrative”. We are happy you found our contributions “compelling”, noting that they add “to the (currently limited) literature on VLM adversarial robustness scaling.” Indeed, we address open questions and clarify when scaling inference compute benefits adversarial robustness. We have revised our submission based on your review and welcome the opportunity to further improve our paper via discussion.
>
> In particular, please let us know if you still believe the language should be softened after accounting for our responses and revision.
>
> ## Questions and Weaknesses
> > It would be nice to have a study where the authors themselves perform adversarial training, and then talk about increasing adversarial robustness as a result of increasing amount of adversarial training … Is there a reason the authors were not able to perform adversarial training themselves?
> >
> > To what extent is it reasonable to say that the three models studied represent low/medium/high amounts of robustness? Is there a way to make this more convincing?
>
> To clarify that the models we selected convincingly possess low, medium, and high robustness levels, we add the following table to our Methodology section, complementing our submission’s discussion of their very different degrees of adversarial training.
>
> | Eval | Model | Vision Encoder | COCO | Flickr30k | VQAv2 | TextVQA | Average |
> |------|-------|----------------|------|-----------|-------|---------|---------|
> | ℓ∞ = 4/255 | LLaVA-v1.5 | OpenAI-L/14 | 3.1 | 1.0 | 0.0 | 0.0 | 1.0 |
> | ℓ∞ = 4/255 | LLaVA-v1.5 | FARE-L/14 | 31.0 | 17.5 | 23.0 | 9.1 | 20.1 |
> | ℓ∞ = 4/255 | Delta2LLaVA-v1.5 | DeltaCLIP-H/14-336 | **95.4** | **57.0** | **61.0** | **32.4** | **61.5** |
>
> We would agree that performing various degrees of adversarial training ourselves, observing the effects of our interventions, is a rigorous approach. However, performing adversarial pretraining of a VLM is out of scope as a very large compute cost that is not necessary when entire papers (i.e., Double Visual Defense) devoted to such efforts have open-sourced their models. Importantly, Delta2LLaVA’s intensive adversarial pretraining (utilizing a week on a TPU v4-512 pod) produced fundamentally different reasoning behaviors on adversarially OOD data, which we might not have observed by doing adversarial training at smaller scales. Such behaviors inspired our paper’s core hypothesis, which we tested rigorously despite not pretraining models.
>
> Addressing the reviewer’s concern, we perform adversarial finetuning instead. While our submission’s experiments found that adversarial finetuning produced a smaller effect size than adversarial pretraining, it nonetheless led to faster robustness gains (as inference compute rose) than having no adversarial training at all, consistent with the RICH. Our revision demonstrates that such adversarial finetuning of InternVL 3.5 gpt-oss 20B leads to a similar enhancement in inference compute’s robustness benefit, producing the first adversarially robust RL-tuned open-weight VLM that we are aware of.
>
> Specifically, more broadly testing our hypothesis, **we leverage the recently released InternVL 3.5 vision language model that uses a gpt-oss LLM to scale reasoning**. We applied FARE, lightweight adversarial finetuning (Schlarmann et al., 2024), to this model’s ViT vision model. Then we applied budget forcing (Muennighoff et al., 2025) to scale the reasoning effort. Results are presented in the revision’s Figure 1 and Appendix D, where we show **scaling reasoning tokens of RL-tuned models via budget forcing improves robustness more if the model is initially more robust, consistent with the RICH.**
>
> Please let us know if we can provide any more details or follow-up experiments. Also, note that we conducted this experiment in the last week, and we expect that we can further improve the finetuning procedure and the resulting model’s performance/robustness in the coming weeks.

---

> > ### Author Response · Authors · 2025-11-21
> >
> > ## Questions and Weaknesses
> >
> > > In Figure 3 (right), you show that increasing K improves robustness across models. But in line 359-361 (in the box), you seem to say that it only helps the already-more-robust model (Delta2). This seems wrong based on Figure 3 (right). It might seem true based on Figure 3 (left), but that's only because the attack saturates, so I don't think it's fair to say even based on that. Would you agree? Am I missing something?
> >
> > Thank you for noting this. We believe we have improved the clarity of this section in the revision, but please let us know if you have questions or uncertainty still. We are happy to make further changes if you have suggestions.
> >
> > Section 4.2 and the box you reference only refer to Figure 3 (left). In the revision, we state that this portion of the figure shows that, at a given PGD step, increasing inference compute has little effect on the attacker’s loss unless the base model is highly robust. We also add a new plot (Figure 3, bottom) that shows the first PGD step on which each model is successfully attacked – this simply visualizes the data of Figure 3 (left) in a new way. This plot clarifies that adding robustness raises the slope of the compute-robustness tradeoff curve – which is fairly flat for models that aren’t Delta2LLaVA in the epsilon 64/255 case – allowing one to obtain more robustness with less compute if the base model is highly robust. The takeaway from this section is that not all models benefit from inference compute equally, some models might appear special.
> >
> > The subsequent section tests whether Delta2LLaVA is indeed uniquely able to gain robustness from inference compute in the presence of white-box attacks. We find that it is not unique: models with lightweight adversarial finetuning (or even no adversarial training) also benefit from inference compute provided the attack is not sufficiently OOD/strong, supporting the RICH.
> >
> > > Figure 1: what happens if you repeat the whole brown text (not just the bit in the curly braces)? Why did you not choose to repeat the whole instruction?
> >
> > Good question! Table R2 below gives the number of PGD steps needed for the attack to succeed when we repeat all of the brown text as you suggested. We observe similar behavior to our submission’s Figure 3 and Table 2.
> >
> > **Table R2** PGD steps required for a successful attack across models, perturbation budget ε, and inference-compute levels K. Computed using a shape attack on the Red Ball where K controls the number of full instruction repetitions.
> >
> > | **Model** | **ε=16/255, K=0** | **ε=16/255, K=1** | **ε=16/255, K=3** | **ε=16/255, K=5** | **ε=64/255, K=0** | **ε=64/255, K=1** | **ε=64/255, K=3** | **ε=64/255, K=5** |
> > | :--- | :---: | :---: | :---: | :---: | :---: | :---: | :---: | :---: |
> > | LLaVA-v1.5 | 5.0 | 9.0 | 14.0 | 10.0 | 7.0 | 10.0 | 9.0 | 9.0 |
> > | FARE-LLaVA-v1.5 | 15.0 | 53.0 | 68.0 | 50.0 | 14.0 | 12.0 | 14.0 | 14.0 |
> > | Delta2LLaVA-v1.5 | Attack Failed | Attack Failed | Attack Failed | Attack Failed | 29.0 | 49.0 | 44.0 | Attack Failed |
> >
> > As for why we chose our approach, note that the security specification in Figure 1 had two components: a general defer instruction (“If the shape in the image conflicts with the following text, defer to the text for your response”) and text that can be compared to the image to determine if there’s a conflict (i.e. “A round soccer ball sits in the grass”). For a model that follows instructions on adversarially OOD data, considering the latter text in the presence of an attacked image (that appears to not be round) would produce a conflict that causes the attack to be resisted (due to the former text), and we hypothesized that we could sharpen that conflict (and thus model robustness to the attack) by just scaling the latter text. Our experiments showed that this is the case, even finding that such scaling caused attackers to produce interpretably more-cuboid soccer balls to overcome the security specification if the model was sufficiently robust (i.e., Delta2LLaVA). Please let us know if this clarification could be improved and/or if it should be added to the text.

---

> ### Author Response · Authors · 2025-11-21
>
> ## Questions and Weaknesses
>
> > in Figure 2, why do you think that adding only a subset of the defenses (middle two columns) leads to *worse* performance (I would have expected it to be at least the same, if not better)?
>
> Briefly, we have a hypothesis that may explain this, and we are happy to look into it further by running tests during the discussion period; please let us know if you have any suggestions related to this. For now, our revision emphasizes only the final two columns, moving the full plot to the appendix, to improve communication of our key message: adding a security specification (and text consistent with its satisfaction) is not sufficient to improve robustness – base model robustness matters.
>
> Hypothesis 1: Adding pre-filling disadvantages the attacker relative to removing pre-filling. Specifically, pre-filling allows the security specification and request for an image description to be simultaneously satisfied if the model simply stops generating text at the end of the pre-fill, which may make the stop token more likely at the expense of the likelihood of the attacker’s target text. Consistent with this, without pre-filling, the security specification added no robustness, yet it was capable of adding robustness with pre-filling (for the model that’s most robust). Without the pre-filling, the lack of a security-specification robustness effect across all models may not be surprising given that this experiment uses a white-box attack to promote the success of a prompt injection, a combined attack much stronger than the black box attacks we and Zaremba et al. (2025) consider.
>
> One way to test this might be adding random tokens to the pre-fill: pre-filling random tokens should not bias the model towards emitting a stop token because the image description task would be incomplete. If we were to observe a robustness benefit of the security specification with such a random pre-fill, this would seemingly contradict the above hypothesis.
>
>
> ## Addressing Suggestions for Improved Clarity
> >  Figure 1 caption: the bold text is not so clear what it's trying to say
>
> This is corrected in the revised manuscript, Section 2. It now reads, “Attacks on models with more base robustness utilize their instruction following, needing increasingly strong visual evidence for the attack target to negate test compute scaling.” Please let us know if more clarity could be provided still.
>
> > line 160: why is there no box around the hypothesis as there is around the questions? seems it should be a fancy box even.
>
> Thank you, we agree and correct this in the revised manuscript in Section 2. Please let us know if you prefer a particular style.
>
>
> > line 214: "If a clear security..." I'm not sure this sentence is true? Like I don't see how it is implied.
>
> Our revision rephrases this line and broadly revises Section 4.1’s introduction – we include a relevant and potentially clarifying excerpt below. Please let us know if your concern remains, we’re happy to add more detail.
>
>
> >>Security specifications are the foundation for test-time compute's robustness effect: they implicitly or explicitly impose a model requirement that inference processes can reference to shift the output probability distribution away from attacker goals. In their adversarial multimodal experiments, \citet{zaremba2025trading} relied on the model's possession of an implicit security specification to disregard minor image perturbations, and they found scaling inference compute could not drive attack success rates towards zero. Here, we consider the possibility that shifting to an explicit security specification will provide robustness benefits, even in not-robustified models.
> >>
> >>Alternatively, robustness benefits of security specifications may depend on and improve with the amount of instruction following and adversarial data in model training data, which could influence compositional generalization to the problem of specification enforcement on adversarially OOD data…

---

> > ### Author Response · Authors · 2025-11-21
> >
> > ## Addressing Suggestions for Improved Clarity
> >
> > > line 215: what does “generalizing to a specification and adversarially OOD data” mean?
> >
> > The revision improves such phrasing throughout the text. For example, the revised Section 2 discusses the point you reference as follows: “Our core argument is that meeting security specifications on adversarially OOD data is more difficult – and perhaps not possible regardless of inference compute scale – without the base robustness needed to follow instructions on such data.” Please let us know if this remains vague.
> >
> > >  all tables: remove vertical bars.
> >
> > Thank you for the suggestion! We addressed this in the revision.
> >
> > > line 297: it's not clear where this question and where its answer come from
> >
> > The revision addresses this, clarifying Section 4.1’s narrative and the takeaway in this box. The box now says, “Can Security Specifications Boost Robustness to Strong Multimodal Attacks? This is possible when using adversarially trained models, which are better equipped to enforce security specifications on adversarially OOD data through compositional generalization.”
> >
> > > table 1: what is the "up arrow" next to the "attacker loss" text?
> >
> > The up arrow indicates that higher attacker loss corresponds to better performance (in many other ML contexts, lower loss is better). Please let us know if we can improve the clarity here further.
> >
> > > table 1: why is there (0.0) next to the Delta2 numbers?
> >
> > This is the standard deviation over several trials of the attack. For Delta2LLaVA models, there was no significant variation in the attacker loss at steps 100 and 300.
> >
> > > table 1: what happens if you don't pre-fill the model response? Please mention in the main text why you needed to do this in order to get it to work.
> >
> > As shown in our submission’s Figure 2, removing pre-filling causes security specifications to provide no benefit. This is perhaps unsurprising given the strength of this white-box attack, which is combined with a prompt injection. Indeed, as discussed above, no model can benefit from security specifications when pre-filling is disabled. Our main text now includes the following, and Appendix B now discusses the above hypothesis regarding disadvantaging the attacker. Could you please let us know if you would like to see more experiments or more commentary in the main text on this point?
> >
> > >> Notably, we pre-fill the model response before the attacker's target text so that the security specification would be satisfied if the model stopped generating output before the attacker's target text; i.e., the model can achieve the goal of describing the image while disregarding the text inside it if it simply assigns low probability to the attacker's target string. See Appendix \ref{app:additional_results} for further discussion and results without pre-filling.
> >
> >
> >
> > > 305: model -> model's
> > >
> > > 415: traditional -> realistic
> >
> > Thank you! We’ve changed the language in these sections to improve readability and correct such errors.

---

### Official Review · Reviewer_5kEV · 2025-10-29

**Soundness:** 2
**Presentation:** 2
**Contribution:** 2
**Rating:** 2
**Confidence:** 4

**Summary:**

This paper investigates whether increasing inference time compute scaling (e.g., security specifications) can systematically enhance model robustness to white-box adversarial attacks in large vision language models. Specifically, the paper hypothesizes that models already robustified via adversarial training can leverage additional inference-time computation (e.g., reasoning, repetition, or Chain-of-Thought prompting) to further resist attacks, while non-robust models cannot. To validate this, the authors conduct a series of white-box and black-box adversarial attack experiments using multiple vision-language models (VLMs) with different robustness levels — including LLaVA-v1.5 (non-robust), FARE-LLaVA (moderately robust), and Delta2LLaVA (strongly robust) — and compare their performance under varying inference-compute settings.

**Strengths:**

- This paper expands on Zaremba et al. (2025) by extending inference-time compute robustness analysis to white-box settings, where attackers have gradient access.
- The results reaffirm the importance of adversarial training during pretraining, showing that such robustness is a prerequisite for inference-time compute to yield benefits under white-box or multimodal attacks.

**Weaknesses:**

- While the paper aims to extend prior work (Zaremba et al. (2025)) by testing inference-time scaling under white-box attacks, this scenario may not be practical in real attack scenarios, as it assumes the attacker has full access to model parameters and gradients — a condition rarely met for attacking state-of-the-art closed-source models. Moreover, in the more practical black-box transfer attack setting, the paper’s results (e.g., Table 3) show that inference-time scaling provides little to no measurable robustness improvement, which weakens the real-world applicability of the proposed hypothesis and its claimed benefits.
- The suggested inference-time scaling methods—such as security specifications or pre-filled responses—may require image-specific ground-truth prior information from the defender, which is impractical and non-scalable in real-world defense settings. Moreover, the paper does not examine how such interventions affect utility on clean images or whether a robustness–utility trade-off exists, leaving the practical impact of these defenses unclear.
- The paper tests its hypothesis on only three models (LLaVA, FARE-LLaVA, and Delta2-LLaVA), which raises concerns about generalizability. With such a narrow set of architectures and training setups, it’s unclear whether the claimed relationship between base robustness and inference-time scaling effects truly holds in broader contexts (e.g., closed-source VLMs, or reasoning-tuned VLMs).
- One of the paper’s central claims —that inference-time compute improves robustness primarily when adversarial inputs resemble the model’s training distribution—is conceptually expected rather than novel. It largely reiterates the established understanding that robustness diminishes as inputs deviate from the training manifold, rather than introducing a fundamentally new mechanism or theoretical perspective.
- Also, the paper claims that even non-robust models can gain robustness from inference-time scaling when facing weaker (i.e., near in-distribution via reduced $\epsilon$) attacks. However, this scenario appears impractical and weakly motivated: in realistic threat scenario, an attacker with full gradient access (as assumed in the white-box setup) would not choose a weak attack. Moreover, as shown in Figure 3 (right), once the reduced $\epsilon$ PGD attack is applied for sufficient steps (≈50), the effect of inference-time scaling nearly vanishes.

**Questions:**

See above weaknesses.

---

> ### Author Response · Authors · 2025-11-21
>
> We are pleased the reviewer found that our work “reaffirm[s] the importance of adversarial training… showing that such robustness is a prerequisite for inference-time compute to yield benefits”. Yes, we diagnose previous inference scaling failures, clarifying the necessary conditions for inference compute scaling’s adversarial robustness benefit. We thank the reviewer for their feedback, hoping they agree that our revised manuscript and comments below address their concerns.
>
> ## Weaknesses
> > While the paper aims to extend prior work (Zaremba et al. (2025)) by testing inference-time scaling under white-box attacks, this scenario may not be practical in real attack scenarios, as it assumes the attacker has full access to model parameters and gradients — a condition rarely met for attacking state-of-the-art closed-source models.
>
> We emphasize that this point was noted in our submission (Line 414) – it was the motivation for the black-box experiments we ran. Critically, while our white-box experiments supported our key hypothesis and clarified how inference compute provides robustness, our black-box experiments are entirely relevant to real attack scenarios and are also consistent with the RICH.
>
> Indeed, we do not agree that our use of white-box experiments is a weakness. Prior work actually called for experiments with stronger attacks like ours: Zaremba et al. (2025) stated that “Further exploration into the adversarial robustness of multimodal models, **including stronger adversarial attacks on images,** remains an interesting direction for future research.”
>
> Zaremba et al. (2025) note in their **white-box language token** experiments that these attacks are unrealistic for deployed frontier models through API access (e.g o1) but perhaps “upper-bound the probability of attacker success." White box access is then a helpful limiting case, and we demonstrate for the first time that inference compute scaling provides multimodal robustness benefits inside it.
>
> Please let us know if including some aspect of the above discussion in the revision would address this weakness in your view, or if the clarifications we provided are sufficient. Thank you!
>
>
>
> > Moreover, in the more practical black-box transfer attack setting, the paper’s results (e.g., Table 3) show that inference-time scaling provides little to no measurable robustness improvement, which weakens the real-world applicability of the proposed hypothesis and its claimed benefits.
>
> There may be a misunderstanding: Table 3, 4 (Table 4, 5 in the revision) show that CoT-based inference compute scaling provides large (~10%) robustness benefits, if and only if the base model is robust. Indeed, we showed that only robustified models (FARE-LLaVA-v1.5 and Delta2LLaVA-v1.5) have statistically significant accuracy improvements on adversarial data after applying CoT.
>
> We also showed that scaling the size of *non-robust* models doesn’t help on adversarial data. The VLMs Qwen-2.5-VL-72B and Llama-3.2-Vision-90B have no statistically significant benefit from CoT on adversarial data, consistent with limited benefits in this setting found by prior work with large-scale non-robust models (Zaremba et al., 2025).
>
> Importantly, all models benefit from CoT on clean data, but only initially-robust models can leverage CoT to improve performance on adversarial data (i.e. add robustness). Thus, these experiments strongly support the RICH and its real-world applicability/benefits.
>
> > The suggested inference-time scaling methods—such as security specifications or pre-filled responses—may require image-specific ground-truth prior information from the defender, which is impractical and non-scalable in real-world defense settings.
>
> Security specifications are practically used in real-world defenses and are commonly discussed in safety research – examples include OpenAI’s deliberative alignment (Guan et al., 2024), Anthopic’s Constitutional AI (Bai et al., 2022), and Meta’s Llama Guard classification (Inan et al., 2023).
>
> We note that security specification practicality can vary, but less practical specifications are useful for testing reasoning’s potential and limitations. Indeed, Zaremba et al. (2025) use such less practical specifications (see Figure 19 in their Appendix 1, which we reproduced/discussed in our submission’s Section 2, paragraph 2).
>
> Also, our submission’s black-box experiments did not rely on explicit or impractical security specifications, and they strongly supported our core hypothesis.
>
> Could you please let us know if adding the above discussion to our revision would address this concern, or if the clarifications suffice? Thank you!

---

> > ### Author Response · Authors · 2025-11-21
> >
> > ## Weaknesses Part II
> >
> > >  Moreover, the paper does not examine how such interventions affect utility on clean images or whether a robustness–utility trade-off exists, leaving the practical impact of these defenses unclear.
> >
> > This is a great point, and we added as a limitation in our revision the note that adding robustness to the base model can come at the cost of clean accuracy. Thank you for raising this point! Perhaps it suggests that our findings are most relevant to safety-specific models (like Llama Guard) rather than models that are meant for clean data.
> >
> >
> > > The paper tests its hypothesis on only three models (LLaVA, FARE-LLaVA, and Delta2-LLaVA), which raises concerns about generalizability. With such a narrow set of architectures and training setups, it’s unclear whether the claimed relationship between base robustness and inference-time scaling effects truly holds in broader contexts (e.g., closed-source VLMs, or reasoning-tuned VLMs)
> >
> > We lack the ability to alter the robustness of closed-source VLMs, but our submission did show that their robustness depends on robust representations (see Appendix A).
> >
> > In our revision, we add testing of RICH in the context of reasoning-tuned VLMs. Our revision demonstrates that the RICH is strongly supported when using InternVL 3.5 (which contains the gpt-oss LLM). We applied FARE, lightweight adversarial finetuning (Schlarmann et al., 2024), to this model’s ViT vision model. Also, we applied budget forcing (Muennighoff et al., 2025) to scale the reasoning effort. Results are presented in the revision’s Figure 1, wherein we show **scaling reasoning tokens of RL-tuned models via budget forcing improves robustness more if the model is initially more robust, consistent with the RICH.**
> >
> > Thus, we have tested six different models, all of which were trained with very different approaches, illustrating the broad support for the RICH. Our submission tested the RICH with 5 models: 3 LLaVA-style models trained with different approaches that induce varying robustness, and 2 large-scale VLMs -- Qwen-2.5-VL-72B and Llama-3.2-Vision-90B.
> >
> >
> > > One of the paper’s central claims —that inference-time compute improves robustness primarily when adversarial inputs resemble the model’s training distribution—is conceptually expected rather than novel.
> >
> > We believe our submission has strong novelty. Could you please let us know if emphasizing any of the following in our revision would improve the communication of our work’s novelty?
> >
> > 1) Zaremba et al. (2025) stated that *“Further exploration into the adversarial robustness of multimodal models, including stronger adversarial attacks on images, remains an interesting direction for future research.”* Our work provides such stronger attacks (white-box). Further, we illustrated improvements to multimodal reasoning’s robustness benefits and provided an experiment-tested hypothesis to explain the improvements’ source, positioning our work to spawn further model security improvements and its own follow-up research
> > 2) Our work offers a critical clarification: while standard LLM pre-training and post-training can produce a model that gains robustness to *text attacks* as inference compute rises, *this does not mean that robustness to vision attacks will also benefit from standard VLM training.* In fact, our work suggests that robustness to some text attacks may come as a free lunch precisely because those text attacks are composed of in-distribution tokens, whereas continuous vision attacks need not be composed of contents seen at training time, harming the potential for compositional generalization. This importantly suggests that inference compute is a flawed defense that can be improved by adversarial training, which we validate. In line with this, we quote recent related work by Nasr et al. (2025) that reacts to the inability of defenses to stop adaptive LLM attacks: *“Only adversarial training that performs robust optimization–where perturbations are optimized inside the training loop–has been shown to yield meaningful robustness (Madry et al., 2018; Athalye et al., 2018) (for the space of attacks used during optimization).”*
> > 3) Our work offers a clear practical benefit: efficiency and thus cost savings. Should one want to defend their LLM/VLM via test-time-compute, e.g. via deliberative alignment, our work suggests that the same robustness can be reached with many fewer reasoning tokens by training the model on the contents of the attacks it will encounter. Alternatively, higher robustness can be reached for the same token cost.
> > 4) Finally, during the rebuttal period, we developed the first (to our knowledge) adversarially robust reasoning VLM by adversarially finetuning the InternVL 3.5 model. Critically, our experiments with this model demonstrate support for the RICH with RL-tuned reasoning VLMs (see Figure 1). We plan to release this model publicly upon paper acceptance, contributing an adversarially robust reasoning VLM to the community.

---

> > > ### Author Response · Authors · 2025-11-21
> > >
> > > ## Weaknesses Part III
> > >
> > > > Also, the paper claims that even non-robust models can gain robustness from inference-time scaling when facing weaker (i.e., near in-distribution via reduced ) attacks. However, this scenario appears impractical and weakly motivated: in realistic threat scenario, an attacker with full gradient access (as assumed in the white-box setup) would not choose a weak attack.
> > >
> > > We emphasize that adversarial robustness papers commonly use epsilons as small as 2/255 – e.g., see “Robust CLIP” (oral at ICML 2024). Moreover, we used white-box attacks with uncommonly high epsilons (64/255) in our experiments, whereas Zaremba et al. (2025) do not even use white-box vision attacks. When we reduced epsilon to 16/255, we were still using a much larger epsilon than what is commonly used in the adversarial robustness literature (e.g. the RobustBench leaderboard from Croce et al., 2020, tracks performance at 4/255).
> > >
> > > More importantly, the goal of reducing epsilon from 64/255 to 16/255 was simply to test the RICH, which suggests that using data closer to the model’s training distribution should enhance inference compute’s robustness benefits. Critically, our experiment found this was the case, supporting the RICH.
> > >
> > > > Moreover, as shown in Figure 3 (right), once the reduced  PGD attack is applied for sufficient steps (≈50), the effect of inference-time scaling nearly vanishes.
> > >
> > > Inference compute is not able to completely prevent white-box PGD attacks, and that is expected – e.g., prior work discusses the ability of white-box attacks to reach attack success rates of 100% (Zaremba et al., 2025). However, we can nonetheless measure robustness benefits of inference compute in white-box settings by counting the number of PGD steps needed for attack success, or by measuring the PGD attacker’s loss at a given PGD step. This is what we do in our white-box experiments, which show that inference compute adds more robustness to models that start off with more initial-robustness.

---

### Official Review · Reviewer_18GW · 2025-10-30

**Soundness:** 3
**Presentation:** 4
**Contribution:** 3
**Rating:** 6
**Confidence:** 4

**Summary:**

This paper proposes the Robustness from Inference Compute Hypothesis (RICH): inference-time compute defenses are most effective when the attacked data's components are closer to the model's in-distribution data. The authors provide empirical evidence across vision-language models, showing that inference compute (e.g., Chain-of-Thought, response pre-filling) significantly boosts robustness primarily for models that are already robust via adversarial training, creating a "rich-get-richer" dynamic.

**Strengths:**

- Novel Conceptual Framework: The RICH hypothesis offers a clear, falsifiable explanation for when inference-compute scaling succeeds or fails for robustness.
- Systematic Evaluation: Rigorous experiments across a matrix of models, attacks, and compute levels.
- Actionable Insight: The synergy between adversarial training and inference compute is a valuable finding.

**Weaknesses:**

- Limited Scope of "Compute": The methods used (prompt repetition, pre-filling) are weak proxies for the complex, multi-step reasoning often implied by "inference compute," limiting the generalizability of the findings.
- Model Scale: Experiments are conducted on smaller VLMs. Validation on frontier-scale reasoning models (e.g., o1-preview) is needed to confirm the hypothesis holds for the most advanced systems.
- Incremental Novelty: The core idea—that generalization benefits from in-distribution data—is foundational. The paper's contribution is applying this to inference-time defense, but the conceptual leap is moderate.

**Questions:**

- Given the reliance on compositional generalization, how do you expect the RICH hypothesis to apply in purely textual domains against sophisticated jailbreaks?
- Is there a point of diminishing returns for inference compute (K) on robust models, or do the benefits continue to scale linearly in your experiments?

---

> ### Author Response · Authors · 2025-11-21
>
> Thank you for your thorough review! We appreciate your feedback and are pleased you found our “novel conceptual framework” to provide a “clear, falsifiable explanation for when inference-compute scaling succeeds or fails for robustness.” Moreover, thank you for identifying our work as providing “actionable insight” backed by “rigorous experiments across a matrix of models, attacks, and compute levels.” Indeed, we believe that the need for “synergy between adversarial training and inference compute is a valuable finding.” Please let us know if we can provide more clarity or results to aid your review.
>
> ## Weaknesses
> >Limited Scope of "Compute": The methods used (prompt repetition, pre-filling) are weak proxies for the complex, multi-step reasoning often implied by "inference compute,"
> >
> >Model Scale: Experiments are conducted on smaller VLMs. Validation on frontier-scale reasoning models (e.g., o1-preview) is needed to confirm the hypothesis holds for the most advanced systems.
>
> While we complemented our prompt repetition and pre-filling experiments with vanilla CoT, we agree that we did not originally provide a strong proxy for RL-instilled complex reasoning. To more fully test our hypothesis and show that the RICH is supported in such settings, **we utilize the recently released InternVL 3.5 vision language model that uses a gpt-oss LLM to scale reasoning**. We applied FARE, lightweight adversarial finetuning (Schlarmann et al., 2024), to this model’s ViT vision model. Also, we applied budget forcing (Muennighoff et al., 2025) to scale the reasoning effort. Results are presented in the revision’s Figure 1, where we show **scaling reasoning tokens of RL-tuned models via budget forcing improves robustness more if the model is initially more robust, consistent with the RICH.**
>
> We believe we have provided comprehensive evidence for the RICH across highly diverse experimental setups. Also, we will retain the limitation that our hypothesis should be verified at frontier compute levels. Please let us know if you recommend any additional limitations or experiments.
>
> > Incremental Novelty: The core idea—that generalization benefits from in-distribution data—is foundational. The paper's contribution is applying this to inference-time defense, but the conceptual leap is moderate.
>
> We believe our submission has strong novelty. Could you please let us know if emphasizing any of the following in our revision would improve the communication of our work’s novelty?
>
> 1) Zaremba et al. (2025) stated that *“Further exploration into the adversarial robustness of multimodal models, including stronger adversarial attacks on images, remains an interesting direction for future research.”* Our work provides such stronger attacks (white-box). Further, we illustrated improvements to multimodal reasoning’s robustness benefits and provided an experiment-tested hypothesis to explain the improvements’ source, positioning our work to spawn further model security improvements and its own follow-up research
> 2) Our work offers a critical clarification: while standard LLM pre-training and post-training can produce a model that gains robustness to *text attacks* as inference compute rises, *this does not mean that robustness to vision attacks will also benefit from standard VLM training.* In fact, our work suggests that robustness to some text attacks may come as a free lunch precisely because those text attacks are composed of in-distribution tokens, whereas continuous vision attacks need not be composed of contents seen at training time, harming the potential for compositional generalization. This importantly suggests that inference compute is a flawed defense that can be improved by adversarial training, which we validate. In line with this, we quote recent related work by Nasr et al. (2025) that reacts to the inability of defenses to stop adaptive LLM attacks: *“Only adversarial training that performs robust optimization–where perturbations are optimized inside the training loop–has been shown to yield meaningful robustness (Madry et al., 2018; Athalye et al., 2018) (for the space of attacks used during optimization).”*
> 3) Next, our work offers a clear practical benefit: efficiency and cost savings. Should one want to defend their LLM/VLM via test-time-compute, e.g. via deliberative alignment, our work suggests that the same robustness can be reached with many fewer tokens by training the model on the contents of the attacks it will encounter. Alternatively, higher robustness can be reached for the same token cost.
> 4) Finally, during the response period, we developed the first (to our knowledge) adversarially robust open-weight reasoning VLM by adversarially finetuning the InternVL 3.5 model. Critically, our experiments with this model show support for the RICH with RL-tuned reasoning VLMs (see Figure 1). We plan to release this model publicly upon paper acceptance, contributing an adversarially robust reasoning VLM to the community.

---

> > ### Author Response · Authors · 2025-11-21
> >
> > ## Questions
> > > Given the reliance on compositional generalization, how do you expect the RICH hypothesis to apply in purely textual domains against sophisticated jailbreaks?
> >
> > The RICH predicts that sophisticated text jailbreaks would weaken inference compute’s ability to add robustness, unless the model had some base robustness against the contents of the sophisticated jailbreaks. This prediction is in line with the fact that defending against human red-teaming and OOD soft-token (continuous) attacks was difficult in Zaremba et al., 2025 (their Section 3.4), while attacks that utilized in-distribution natural language and simpler strategies that may look more like the contents of the training data were better defended against. Broadly, the view that inference compute works by compositional generalization is consistent with the idea that robustness to sophisticated attacks requires sampling from the space of such attacks to teach the model how to avoid them, which is expressed in full in the quotation provided above.
> >
> > We would be happy to integrate this discussion into a future work or limitations section in our revision – could you please let us know if one of those sounds appropriate to you?
> >
> > > Is there a point of diminishing returns for inference compute (K) on robust models, or do the benefits continue to scale linearly in your experiments?
> >
> > This is an interesting question because negative or diminishing returns from scaling context window size/contents could be due to a model’s limited ability to work with large contexts effectively, or from limitations of security specifications. Avoiding this confound, our new experiments use a model trained to do long-context reasoning – InternVL 3.5 with gpt-oss – and find that robustness benefits persist out to 16K tokens, then level or fall off by 64K tokens (revision's Figure 7). This suggests that large-scale inference compute scaling to meet security specifications can continue to provide benefits, and that diminishing returns may largely come from limited ability to operate on very long contexts.
> >
> > Indeed, our original submission showed LLaVA model robustness benefits plateau when inference compute is scaled sufficiently. See Table 2 (revision's Table 3), where increasing the number of clean image descriptions from K=3 to K=5 does not delay attack success at either perturbation budget. However, similar to our InternVL 3.5 results, Delta2LLaVA-v1.5 continued to show benefits when increasing K from 3 to 5, suggesting a combination of model long-context abilities and base robustness determine reasoning’s robustness benefit, consistent with the RICH.

---

> > ### Comment · Reviewer_18GW · 2025-11-22
> >
> > Thank you for the thorough and constructive rebuttal. The authors have directly addressed my primary concerns.
> >
> > The new experiments with the adversarially fine-tuned InternVL 3.5 model are a particularly strong addition, providing a much more compelling proxy for complex, multi-step reasoning and directly countering the "Limited Scope" weakness. The discussion on diminishing returns and the clarification of the paper's novelty are also effective.
> >
> > Based on these significant enhancements, I have raised my score to reflect the strengthened contribution and validation of the RICH hypothesis.

---

> > > ### Author Response · Authors · 2025-12-03
> > >
> > > It's great to hear that our rebuttal's "significant enhancements" directly addressed your primary concerns. Thank you for your feedback that improved the submission and for raising your score!
> > >
> > > Authors of 14637

---

### Official Review · Reviewer_rpuG · 2025-11-05

**Soundness:** 3
**Presentation:** 3
**Contribution:** 2
**Rating:** 6
**Confidence:** 3

**Summary:**

This paper proposes the Robustness from Inference Compute Hypothesis: that the closer an attack's contents are to being in-distribution for the model, the more inference-time compute scaling improves robustness. It then suggests that robustified models can defend against white-box or multi-modal attacks, since the adversarial attacks are more within distribution for the model. It studies this in the vision language model setting with empirical experiments. The paper uses various LLaVA-based models with increasing amount of adversarial training, and finds that using more inference scaling in the form of adding security specifications or pre-filling improves robustness, as well as CoT-prompted VLM models.

**Strengths:**

Originality: The paper proposes the RICH theory for explaining when adversarial robustness benefits from inference compute scaling.
Quality: The main claims of the paper are supported by the experiments
Clarity: The paper is generally clear and well-written, with separately broken out hypotheses and research questions.
Significance:
* Addresses adversarial robustness in VLMs, which has unique challenges compared to other settings
* The paper addresses an open question in from Zaremba et al. (2025)

**Weaknesses:**

Clarity:

W1) I needed to read over things a few times to shift gears between thinking about whether or not the data is in-distribution for the model, to whether or not the data is in-distribution for _following the security specification_. I think I now understand that the paper has a large focus on confirming that adversarially trained models are more able to generalize their instruction following training to enforcing the security specification on adversarial inputs.

W2) The story around prefilling is less fleshed out than the story around the security specification.

**Questions:**

My main questions are around supporting the specific details of the RICH hypothesis more thoroughly.

Q1) How are the experiments different from the null hypothesis that models which have undergone more adversarial training are more adversarially robust?

Q2) Data can be more or less in-distribution even without adding adversarial attacks, for instance by choosing a particular domain not well represented in the training data, or by adding non-adversarial noise or transformations to an image. Are there results which suggest that modifying _in-distribution-ness_ itself helps to modulate adversarial robustness for robustified and non-robust models, besides varying the number of PGD steps in Table 2?

Q3) Inference time scaling in the form of using more CoT tokens rather than more repetitions of externally created text seems to be fairly different. Is there a relationship between CoT length and adversarial robustness? The main experiment that investigates inference time scaling appears to be in Table 2.

Q4) What's the relationship between the prefill and inference compute scaling? It's unclear to me if the paper explored repeating the prefill or not.

---

> ### Author Response · Authors · 2025-11-21
>
> Thank you for your thoughtful review! We are pleased that you believe we address “an open question” in VLM adversarial robustness by proposing “the RICH theory for explaining when adversarial robustness benefits from inference compute scaling.” Indeed, we propose the RICH to understand limitations of scaling inference compute noted by Zaremba et al. (2025): we emphasize that test-time scaling’s robustness benefit requires not just a (possibly implicit) security specification, instead showing that the ability to follow such a specification on adversarially OOD data is granted by adversarial training. We consistently observe that the robustness benefits of inference time compute scaling are reenabled by adversarial training, per the RICH’s prediction. We are glad you found our work “generally clear and well-written”, and the following addresses your comments to enhance its presentation and contributions.
>
> ## Weaknesses
> > I needed to read over things a few times to shift gears between thinking about whether or not the data is in-distribution for the model, to whether or not the data is in-distribution for following the security specification. I think I now understand that the paper has a large focus on confirming that adversarially trained models are more able to generalize their instruction following training to enforcing the security specification on adversarial inputs.
>
> Exactly, your understanding is correct, and we rewrote our submission to make this specific point more clear. Specifically, our revision’s Section 2 discusses this topic (in paragraphs 4-7), clarifying “Our core argument is that meeting security specifications [both explicit and implicit] on adversarially OOD data is more difficult – and perhaps not possible regardless of inference compute scale – without the base robustness needed to follow instructions on such data. At the same time, adding base robustness can enable the instruction following needed to meet security specifications and thereby enhance the robustness benefits inference compute provides.”
>
> Please let us know if we can add anything else to aid the clarity of this critical point.
>
> > The story around prefilling is less fleshed out than the story around the security specification.
>
> The revised manuscript improves the clarity of this story, clearly defining its role in relation to the manuscript's core narrative and hypothesis. Now, at the start of the revision's Section 4 (Line 234), before any experiments are discussed, we clearly explain each experimental subsection and its motivation. For example, we state that:
>
> *"Section 4.1 finds that a security specification is not sufficient to deter attacks, model training must grant the ability to enforce the specification in their presence (e.g., via compositional generalization). Section 4.2 reveals that naively scaling inference compute enhances robustness further, in accordance with the RICH..."*
>
> We then provide more detail in each subsection. For example, in the revised Section 4.1's introduction (Line 249), we state that:
>
> *"Security specifications are the foundation for test-time compute’s robustness effect: they implicitly or explicitly impose a model requirement that inference processes can reference to shift the output probability distribution away from attacker goals."*
>
> Thus, as we also discuss in our response to your final question (below), Section 4.1 is not about inference compute scaling, it is about whether security specifications can be effective at all in the presence of strong white-box attacks. **The null hypothesis is that adding a security specification will not improve the robustness of VLMs to strong white-box attacks, or at least that such a specification would provide a similar benefit regardless of the model's level of adversarial training. We reject the null, finding that a security specification provides a benefit only if the model is robust.** Building on this, Section 4.2 then begins by stating:
>
> *"Given the ability to obtain robustness benefits from security specifications on data affected by white-box multimodal attacks, we now consider whether scaling inference compute enhances the robustness benefit of the security specification, as it did in Zaremba et al. (2025) with weaker attacks."*
>
> We provide further pre-filling clarifications in our response to your related question below, but please let us know if we can add anything else to the revision or to this discussion that might help. Thank you!

---

> ### Author Response · Authors · 2025-11-21
>
> ## Questions
> > How are the experiments different from the null hypothesis that models which have undergone more adversarial training are more adversarially robust?
>
> This is a question we have gotten a couple of times, and we believe our revised text – e.g. the Results and discussion component of Section 4.2 – and new results provide clarification. Please let us know if the following does not address your question well.
>
> Models that undergo more adversarial training start off more robust, but the following was unclear: **”Does this robustness gap grow as inference compute is applied to satisfy security specifications?”. The null hypothesis is no**. In other words, we asked, is the slope of the compute-robustness plot altered by the initial robustness level? Our revision’s Figure 4 (bottom) shows this slope change, and across various other experiments we see broad support for our alternative hypothesis (RICH), clarifying how inference compute provides a robustness benefit in an actionable manner.
>
> Our revision’s Figure 1 makes this point using a vision language reasoning model (InternVL 3.5 with a gpt-oss LLM), showing that a small initial robustness difference grows as the model reasons. Note that this experiment emulates the adversarial vision experiment of Zaremba et al. (2025) – testing on the same dataset and also using an LLM trained to reason with RL – while aligning with results from our original experiments, finding again that reasoning’s robustness benefit is enhanced when starting off with a more robust model.
>
> In other words, it was not previously clear if “the rich get richer” as compute is spent. The RICH predicts that this will happen, and our experiments demonstrate that it does indeed happen.
>
>  > Are there results which suggest that modifying in-distribution-ness itself helps to modulate adversarial robustness for robustified and non-robust models, besides varying the number of PGD steps in Table 2?
>
> Our understanding is that you mean PGD epsilon, not steps, as we used the former to alter in-distribution-ness.
>
> This is an interesting question, and we would be happy to run an experiment in our setup if you have a particular dataset/transformation or hypothesis in mind. Notably, there exists prior work on interactions between OOD robustness and adversarial robustness: e.g., Kireev et al., 2022 -- "On the effectiveness of adversarial training against common corruptions" -- show that appropriately configured adversarial training can help performance on some OOD shifts.
>
> >  Inference time scaling in the form of using more CoT tokens rather than more repetitions of externally created text seems to be fairly different. Is there a relationship between CoT length and adversarial robustness? The main experiment that investigates inference time scaling appears to be in Table 2
>
> Indeed, our experiments utilized externally created text to illustrate that instruction following on adversarially OOD data, not the mere presence of a security specification, mediates inference compute’s robustness benefits. Additionally, it is correct that we went beyond these settings to apply our hypothesis to CoT experiments, which more closely resembled those of Zaremba et al. (2025), matching the dataset they used but lacking a vision language model trained with RL to reason like o1-v.
>
> To more fully test our hypothesis and show that the RICH is supported in settings with scaled reasoning, **we utilize the recently released InternVL 3.5 vision language model that uses a gpt-oss LLM to scale reasoning**. We applied FARE, lightweight adversarial finetuning (Schlarmann et al., 2024), to this model’s ViT vision model. Finally, we applied budget forcing (Muennighoff et al., 2025) to scale the reasoning effort. Results are presented in the revision’s Figure 1, where we show **scaling reasoning tokens of RL-tuned models via budget forcing improves robustness more if the model is initially more robust, consistent with the RICH.** To our knowledge, this adversarially trained InternVL 3.5 is the first open-weight reasoning-capable VLM with demonstrated adversarial robustness, which we plan to release publicly upon acceptance.
>
> > What's the relationship between the prefill and inference compute scaling? It's unclear to me if the paper explored repeating the prefill or not.
>
> Please see our response to the related weakness above, wherein we clarify this relationship. In sum, Section 4.1 shows that the instruction following that we pre-fill is superficial for non-robust models: attacks easily go around the only-superficially-compliant contents of the context window. Correspondingly, we would not expect robustness benefits of scaling inference compute in order to satisfy a security specification -- because we saw attacks were unaffected by a security specification satisfied by pre-filling in Section 4.1 -- unless the model was somewhat robust. Subsequent sections, like Section 4.2, support this perspective and thus the RICH.

---

### Author Response · Authors · 2025-12-03
**Final Author Remarks**

We thank the reviewers, original AC, and new AC for their efforts during the review process. We are encouraged by the broadly positive reviewer response to our original submission, the helpful suggestions from reviewers, and the positive response to the changes we added during the rebuttal.

Indeed, our rebuttal’s “significant enhancements” “directly addressed” [18GW] reviewer concerns. For example, “a particularly strong addition” [18GW] was showing that our core hypothesis (RICH) is critical to performance in a practical and timely black-box-attack setting. **To demonstrate this, we trained -- and are prepared to release -- the first open-weight adversarially robust reasoning VLM:** a robustified InternVL 3.5 gpt-oss 20B created by adversarially finetuning the vision encoder ViT. As our new Figure 1 shows, this new model gains significant additional robustness when we increase gpt-oss 20B reasoning duration (via budget-forcing), whereas the base model does not. **This provides strong empirical support that the RICH matters in practice.**

Moreover, we addressed all reviewer concerns regarding our experiments with stronger white-box attacks that rigorously test the RICH. For instance, we improved the introduction to our prompt injection experiments, clarifying their connection to the manuscript’s core narrative. **We also added ablation studies (Appendices B and C) showing that inference-scaling defenses with random tokens do not provide a benefit.** Instead, as predicted by the RICH, test time compute provides robustness benefits -- even on strong white-box, prompt-injection attacks -- provided that (1) there is a security specification that instructs the model to resist attacks; and (2) the model has enough base robustness to follow instructions on adversarially OOD data, allowing the security specification’s attack-thwarting goal to influence the model’s text-generation probabilities at test time.

Thank you,

Authors of 14637

---

### Meta-Review · Area_Chair_B5RY · 2026-01-06

**Summary:**

The submission received divergent ratings, including 3 positive scores and 1 negative score.

Reviewer rpuG mainly critiques clarity and raises several questions about validating the RICH hypothesis. Reviewer 18GW raised concerns about the technical novelty and expected to see more experiments on model scaling. Reviewer 5kEV gives the most critical review and argues that the setting of white-box attacks is not practical. In the meantime, it is also pointed out that testing its hypothesis on only three models is insufficient, and the core claims are unsurprising. Reviewer Xw6j finds "RICH" novel while also acknowledging that the results are not groundbreaking.

**Reviewer Concerns:**

The authors have provided a detailed response to address the comments one-by-one. Reviewer 18GW has been successfully engaged with the discussion and tends to raise the score. The AC devoted much more effort to the Reviewer 5kEV's comments and the corresponding response, and believes that the authors have provided an adequate rebuttal, which would assuage the concern. However, the AC agrees with Reviewer rpuG that the clarity should be improved to better highlight the focus of this work.

**Reviewer Scores:**

Reviewer rpuG and Reviewer Xw6j keep the positive rating.

Reviewer 5kEV and Reviewer 18GW increase the rating.

---

### Decision · Program_Chairs · 2026-01-26

Accept (Poster)